# Differentiation dynamics of mammary epithelial cells revealed by single-cell RNA sequencing

Karsten Bach[1,2,3], Sara Pensa[1,3], Marta Grzelak[2,3], James Hadfield [2,3], David J. Adams[3,4], John C. Marioni[2,3,4,5] & Walid T. Khaled[1,3]

Characterising the hierarchy of mammary epithelial cells (MECs) and how they are regulated during adult development is important for understanding how breast cancer arises. Here we report the use of single-cell RNA sequencing to determine the gene expression profile of MECs across four developmental stages; nulliparous, mid gestation, lactation and post involution. Our analysis of 23,184 cells identifies 15 clusters, few of which could be fully characterised by a single marker gene. We argue instead that the epithelial cells—especially in the luminal compartment—should rather be conceptualised as being part of a continuous spectrum of differentiation. Furthermore, our data support the existence of a common luminal progenitor cell giving rise to intermediate, restricted alveolar and hormone-sensing progenitors. This luminal progenitor compartment undergoes transcriptional changes in response to a full pregnancy, lactation and involution. In summary, our results provide a global, unbiased view of adult mammary gland development.

[1] Department of Pharmacology, University of Cambridge, Cambridge CB2 1PD, UK. [2] Cancer Research UK Cambridge Institute, University of Cambridge, Cambridge CB2 0RE, UK. [3] Cancer Research UK Cambridge Cancer Centre, Cambridge CB2 0RE, UK. [4] Wellcome Trust Sanger Institute, Wellcome Genome Campus, Hinxton, Cambridge CB10 1HH, UK. [5] European Bioinformatics Institute, European Molecular Biology Laboratory, Hinxton CB10 1SD, UK. Correspondence and requests for materials should be addressed to J.C.M. (email: marioni@ebi.ac.uk) or to W.T.K. (email: wtk22@cam.ac.uk)

The purpose of the mammary gland is to provide nourishment and passive immunity for the young until they are capable of feeding themselves. From a developmental biology perspective, the mammary gland is a unique organ as it undergoes most of its development during puberty and adulthood[1–4]. In the pre-pubertal mouse, the mammary gland consists of a rudimentary epithelial ductal structure embedded within a mammary fat pad, which is connected to the nipple[5, 6]. At the onset of puberty and in response to hormonal changes, the rudimentary ductal structure will proliferate and migrate to fill the entire mammary fat pad, leaving a developed network of ductal structures that later serve as channels for milk transport during lactation. At the onset of pregnancy,

a highly proliferative stage is initiated, characterised by further ductal side-branching and widespread lobuloalveolar development[1]. Differentiation of the epithelial cells within alveoli prepares the gland for milk production and secretion. Towards the end of pregnancy, the gland is extremely dense and primarily occupied by epithelial cells and very little fat. This morphology is largely maintained throughout lactation. However, in response to cessation of suckling the gland undergoes involution, which is characterised by extensive cell death and tissue remodelling[4, 7]. Towards the end of involution, the gland reaches a morphology resembling that of glands prior to pregnancy and subsequent pregnancies will trigger the same chain of events.

**Fig. 1** Single-cell RNA sequencing identifies 15 clusters of mammary epithelial cells. **a** Schematic diagram highlighting the experimental setup for isolating and sequencing the RNA of single cells using the 10× chromium system. **b** t-SNE plot of 23,184 cells visualises general structure in the data. Cells are coloured by the four developmental time points as follows: pink = NP, dark green = G, light green = L, purple = PI. **c** Same as **b** but colouring cells by clusters. **d** t-SNEs coloured by the normalised log-transformed expression of the basal marker *Krt5* and the luminal marker *Krt18*

Recent efforts have focused on the identification and characterisation of the various mammary epithelial cell lineages within the gland that contribute to this developmental homoeostasis. Pioneering fat pad transplantation studies nearly 70 years ago were the first to demonstrate the regenerative and differentiation capacity of small numbers of cells[8–10]. More recently the use of cell surface markers coupled with flow cytometry has been used to enrich for various progenitor and stem cell compartments[10–13] and showed that imbalance of such cell populations results in cellular transformation and subsequently breast cancer[14, 15]. Other studies, inspired by breast cancer transcriptomic profiling, have identified transcriptional regulators of mammary epithelial cell types such as *GATA3* in luminal cells[13, 16]. More recently, elegant lineage-tracing studies used key markers to address the contribution of each lineage to adult mammary epithelial cell homoeostasis[4]. However, in all of these studies only a handful of markers and genes were used to define the cellular hierarchy of the mammary epithelial cells, with a principal focus on the nulliparous developmental stage. Therefore, to properly understand its changing role throughout life, there is a need for an unbiased and comprehensive characterisation of mammary epithelial cell compartments at different developmental stages.

Here we used single-cell RNA sequencing (scRNAseq) to map the cellular dynamics of mammary epithelial cells across four adult developmental stages; nulliparous, mid gestation, lactation and post weaning (full natural involution). Our data from 23,184 individual cells identify 15 distinct cell populations within the

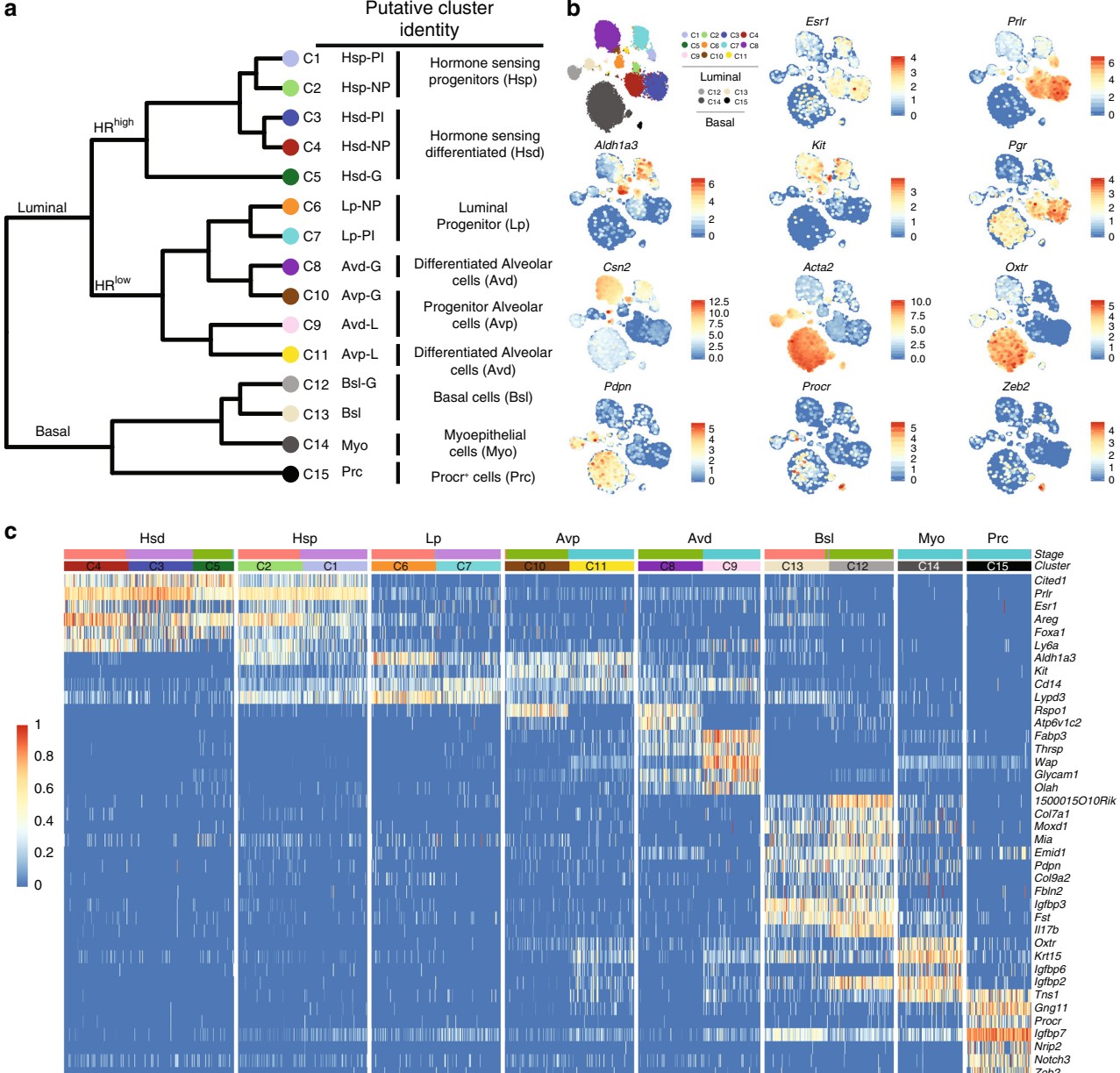

**Fig. 2** Putative identities of mammary epithelial cell clusters. **a** Dendrogram of clusters based on the log-transformed mean expression values of the 15 clusters. The tree was computed based on Spearman's rank correlation with Ward linkage. **b** t-SNEs with overlaid expression of cluster-specific genes. **c** Heatmap highlighting key marker genes that were used to infer putative identities. Colour scale represents log-transformed and normalised counts scaled to a maximum of 1 per row. Upper bars represent the cluster assignment and stages for the individual cells. For visualisation purposes only 100 randomly selected cells were shown for large clusters

**Table 1 Summary of mammary epithelial cell clusters**

| Cluster | Key genes | Number of cells captured | | | | | | | | Putative identity | Name |
|---|---|---|---|---|---|---|---|---|---|---|---|
| | | NP1 | NP2 | G1 | G2 | L1 | L2 | PI1 | PI2 | | |
| C1 | *Esr1, Prlr, Pgr, S100a6, Cited1,* | 1 | 0 | 0 | 0 | 0 | 0 | 107 | 385 | Hormone sensing | Hsp-PI |
| C2 | *Aldh1a3, Cd14* and *Kit* | 265 | 169 | 0 | 0 | 0 | 0 | 5 | 2 | progenitors | Hsp-NP |
| C3 | *Esr1, Prlr, Pgr, S100a6* and *Cited1* | 12 | 5 | 0 | 0 | 0 | 0 | 412 | 2487 | Hormone sensing | Hsd-PI |
| C4 | | 971 | 1212 | 0 | 0 | 0 | 1 | 40 | 88 | differentiated | Hsd-NP |
| C5 | | 0 | 0 | 20 | 41 | 2 | 0 | 0 | 0 | | Hsd-G |
| C6 | *Aldh1a3, Cd14* and *Kit* | 372 | 316 | 2 | 3 | 0 | 2 | 22 | 9 | Luminal progenitor | Lp-NP |
| C7 | | 0 | 1 | 0 | 0 | 0 | 0 | 824 | 1102 | | Lp-PI |
| C8 | *Wap, Csn2, Glycam1* and *Lalba* | 0 | 0 | 1926 | 1818 | 1 | 1 | 0 | 1 | Alveolar differentiated cells | Avd-G |
| C9 | | 0 | 0 | 0 | 0 | 42 | 47 | 0 | 0 | | Avd-L |
| C10 | *Wap, Csn2, Glycam1, Lalba, Aldh1a3,* | 2 | 1 | 89 | 126 | 3 | 2 | 0 | 1 | Alveoloar progenitor cells | Avp-G |
| C11 | *Cd14* and *Kit* | 0 | 0 | 0 | 0 | 142 | 89 | 0 | 0 | | Avp-L |
| C12 | *Krt4, Krt14, Pdpn, Etv5* and *Acta2* | 504 | 282 | 2 | 10 | 1 | 1 | 27 | 53 | Basal cells | Bsl-G |
| C13 | | 0 | 1 | 525 | 594 | 1 | 0 | 0 | 5 | | Bsl |
| C14 | *Oxtr, Acta2, Krt4* and*Krt14* | 0 | 0 | 1 | 0 | 4637 | 3104 | 1 | 1 | Myoepithelial cells | Myo |
| C15 | *Procr, Igfbp4, Gng11* and *Zeb2* | 0 | 1 | 0 | 0 | 205 | 57 | 2 | 0 | Procr + basal cells | Prc |

Overview of the different clusters including number of cells captured for each time-point and key genes that were used to infer their identities

gland and allow their hierarchical structure across developmental time points to be charted.

## Results

**Single-cell RNA sequencing identifies 15 clusters of mammary epithelial cells**. We isolated mammary epithelial cells from four developmental time points; nulliparous (NP), day 14.5 gestation (G), day 6 lactation (L) and 11 days post natural involution (PI). For each time point, we sorted mammary epithelial cells based on the EpCAM cell surface marker from two independent mice (Supplementary Fig. 1; Fig. 1a). All samples were then prepared for scRNAseq using the 10× Chromium platform[17]. Following quality control (Methods), this yielded an average of 6175 unique transcripts and 2118 genes detected from 25,010 cells (4223 in NP, 5826 in G, 9319 in L and 5642 in P) with high reproducibility between the biological replicates (Supplementary Figs. 2 and 3). Visual inspection of the data using t-distributed stochastic neighbour embedding (t-SNE) suggested that although there is grouping of cells by time point, there are also other factors that underlie structure within the data set (Fig. 1b). First, we dissected the global structure by unsupervised clustering using a shared-nearest-neighbour clustering approach (see Methods). This resulted in a coarse clustering into 13 groups (Supplementary Fig. 4a). In a second step, we applied hierarchical clustering to each of the identified groups to further resolve the cellular heterogeneity (Fig. 1c, Supplementary Fig. 4b). After removal of endothelial and immune cell clusters (Supplementary Fig. 4c–e), this resulted in a total of 15 mammary epithelial cell clusters (C1–C15) of 23,184 cells. Based on the expression of *Krt18, Krt8, Krt5, Krt14* and *Acta2*, we noted that 11 clusters showed a luminal profile (C1–C11) and 4 showed a basal profile (C12–C15) (Fig. 1d; Supplementary Data 1–15).

To further characterise the clusters, we identified differentially expressed genes and inferred putative identities based on known marker (Fig. 2a, b). As shown in Table 1, multiple clusters of cells have been assigned to the same putative cell type. We found that in these instances the cells showed a high similarity and expressed the same marker genes but derived from different developmental stages (see NP, G, L or PI suffix in Fig. 2a), suggesting that the

stage of the gland has specific effects on the transcriptional landscape of certain cell types.

In the luminal compartment, we found two larger subgroups (Fig. 2a). C1–C5 showed characteristics of hormone-sensing cells (Hs) such as high expression levels of hormone receptors (*Pgr, Esr1, Prlr*) (Fig. 2b, c). Of the putative Hs cells, we noted that C1/C2 expressed progenitor markers (e.g. *Aldh1a3, Cd14, Kit*)[18], suggesting restricted progenitor function (Hsp) (Fig. 2b, c), whereas C3/4/5 showed a more differentiated state (Hsd) (Fig. 2b, c; Table 1). The second subgroup within the luminal compartment (C6–C11) expressed only low levels of hormone receptors. C6/7 expressed high levels of progenitor markers, which is consistent with a luminal progenitor (Lp) phenotype (Fig. 2b, c). In contrast, C8–C11 expressed milk proteins (e.g. *Wap, Csn2*), and were exclusively composed of cells from gestation and lactation, supporting the hypothesis that these are secretory alveolar cells (Av) (Fig. 2b, c; Table 1). Of the putative alveolar cells, we noted that C10/C11 expressed genes associated with a restricted alveolar progenitor function (Avp), whereas C8/C9 appeared to be in a more differentiated alveolar state (Avd) (Fig. 2b, c).

In the basal compartment we found that cells from C14 showed characteristics of myoepithelial cells such as high levels of *Acta2, Oxtr* and *Krt15* (Fig. 2b, c). In addition, the basal compartment also contained a cluster of cells, C15, that expressed high levels of *Procr, Gng11* and *Zeb2* suggesting that these represent the previously identified Procr+ basal cells, which have been shown to be enriched for mammary repopulating units[19] (Fig. 2b, c).

**Reconstruction of the luminal differentiation hierarchy**. We focused on cells from the NP and G time points to investigate mammary epithelial differentiation states of the gland. These differentiation states and the transitions between them can be computationally reconstructed using diffusion maps[20]. Briefly, the method embeds the data in a low-dimensional space, where distances between cells represent the progression through a gradual but stochastic process such as differentiation. In diffusion maps constructed from all epithelial cells, we observed a clear segregation between the luminal and basal clusters, with virtually

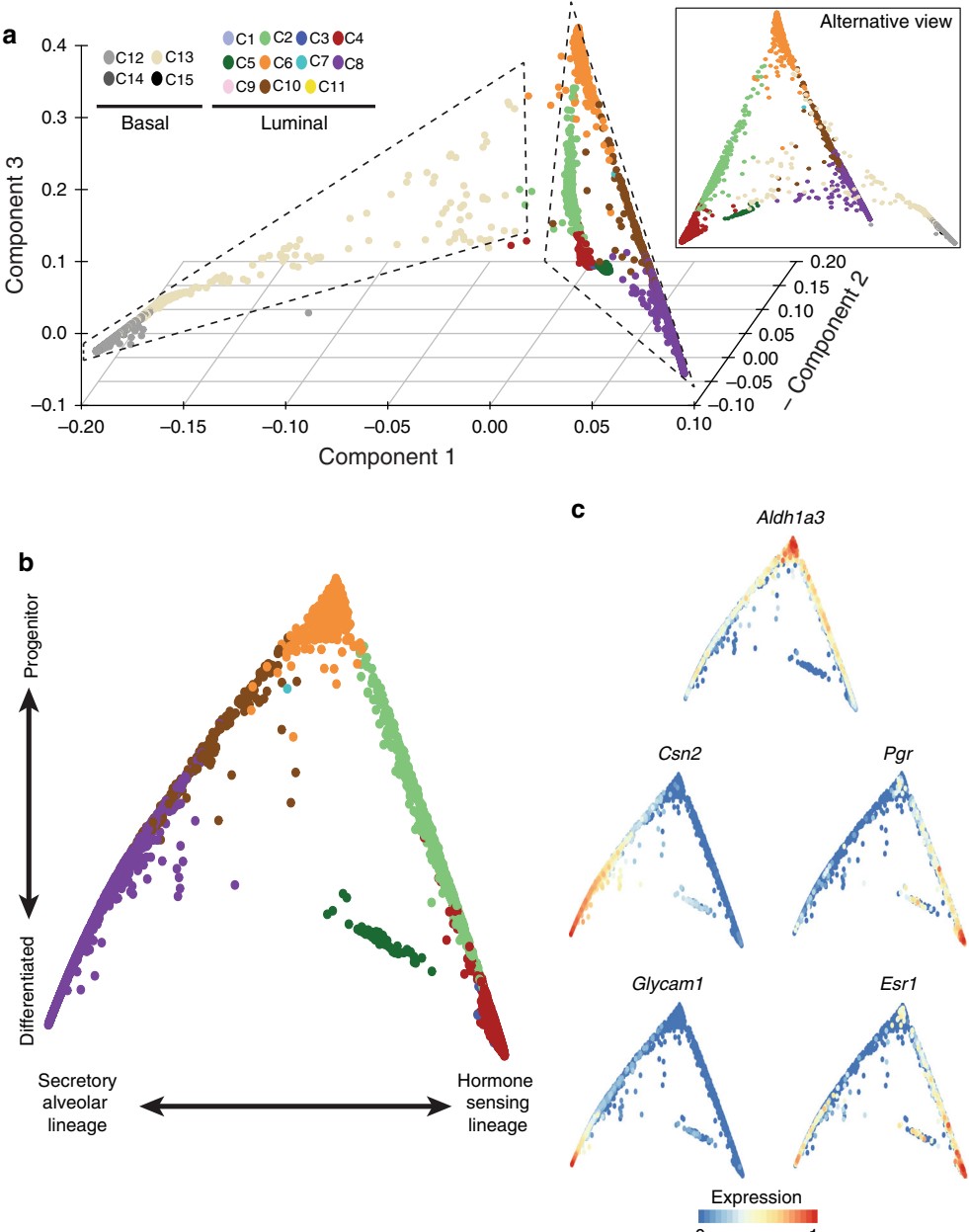

**Fig. 3** Computational reconstruction of differentiation processes in the mammary gland. **a** Diffusion map of epithelial cells from the NP and G time points, showing the first three diffusion components. **b** Differentiation trajectory of the luminal compartment based on the first two diffusion components. **c** Same plot as in **b**, coloured by the normalised and scaled expression values of various genes

no transition states between the two (Fig. 3a). This supports the hypothesis that during normal tissue homoeostasis the two lineages are largely self-maintained, which is in agreement with the majority of lineage tracing studies[21–23]. In contrast, the luminal compartment showed a distinct structure, with gradual transitions between different clusters and cells originating from a common origin (Fig. 3b). We confirmed the robustness of this bifurcation by verifying that it was present when different methods of feature selection, algorithms for trajectory inference and down-sampling were employed (Supplementary Fig. 5).

We found the expression of the progenitor marker *Aldh1a3* gradually decreased as cells progressed away from their common origin (Fig. 3c), which was largely composed of cells from C6 (Lp). We further noted that the left arm of the differentiation trajectory terminates at C8 (Avd) and shows increasing expression of *Csn2* and *Glycam1* (Fig. 3c), consistent with a secretory

phenotype. In between C6 and C8, we found cells that were mainly derived from C10 (Avp, Fig. 3b). On the right arm of the differentiation trajectory, cells from C6 transitioned to C2 and C4/5 (Hsp and Hsd), during which the expression of *Esr1* and *Pgr* increased suggesting that this branch represents differentiation towards hormone-sensing luminal cells (Fig. 3c).

Being confident that the diffusion map recapitulates the luminal differentiation process, we then computationally inferred the two branches and ordered the cells according to their progression through 'pseudotime'[24] (see Methods for further explanation) (Fig. 4a). This allows us to identify genes whose expression changes during the process of differentiation. We found 456 genes that showed a pseudotime-dependent expression with the same directionality along both differentiation trajectories (see Methods for definition, Supplementary Data 16). These included genes associated with known progenitor characteristics

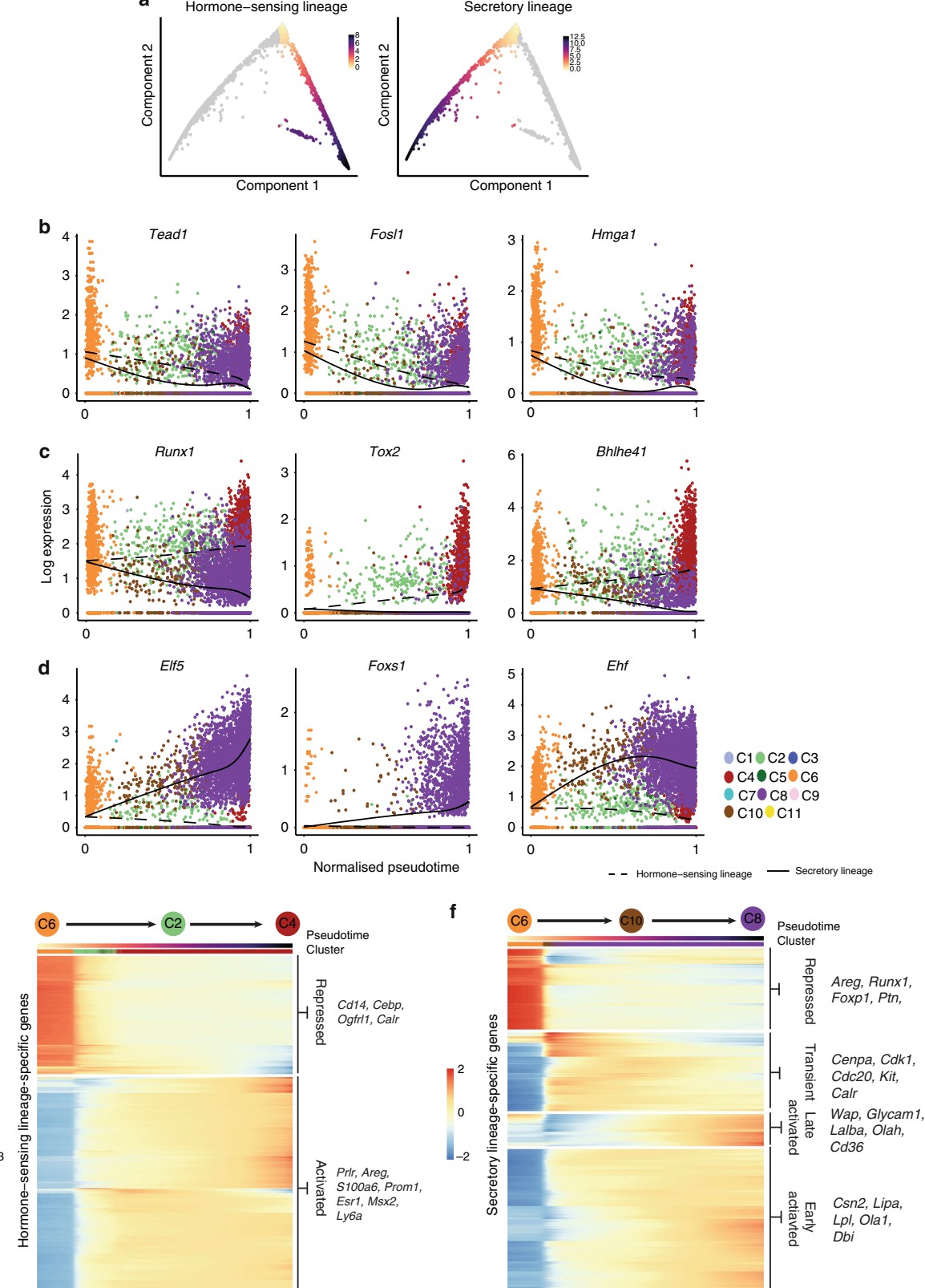

**Fig. 4** Pseudotime ordering identifies genes associated with luminal differentiation. **a** Definition of the hormone-sensing and secretory differentiation branch. Cells are coloured by their progression through pseudotime, where low values represent undifferentiated cells. **b–d** Examples of transcription factors with pseudotime-dependent expression with the same overall trend on both branches (**b**) or branch specific trends (**c**, **d**). **e**, **f** Heatmap of all genes with branch-specific, pseudotime-dependent expression for the hormone-sensing lineage (**e**) or the secretory lineage (**f**). Pseudotime and the cluster assignment are annotated above the heatmap. The values in the heatmap represent z-scaled, spline-smoothed expression values. Genes in the heatmaps were clustered using hierarchical clustering with a dynamic tree cut

such as *Aldh1a3* and *Tspan8*[25] as well as transcription factors that have not previously been associated with luminal differentiation such as *Creb5*, *Hmga1* and *Fosl1* (Fig. 4b). *Hmga1* is a known chromatin remodeller with reprogramming activity that has been

implicated in basal-like breast cancer[26, 27]. In addition, we identified 1005 genes with branch-specific expression patterns. We then clustered the gene expression trend on each of the two branches and identified sets of genes that change over pseudotime

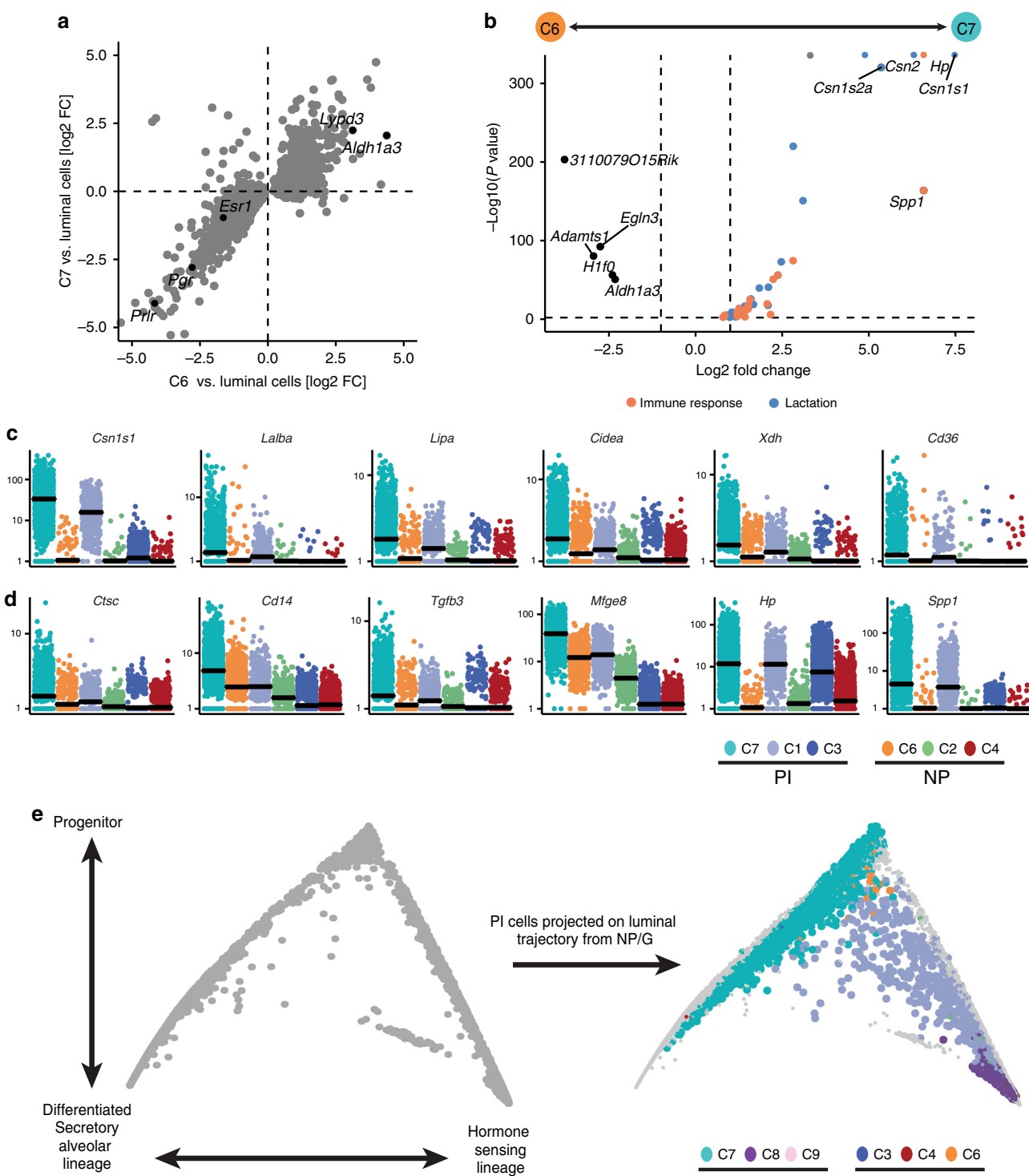

**Fig. 5** The effect of parity on the transcriptomic landscape of the luminal progenitor compartment. **a** Comparison of fold changes from C6 vs. luminal compartment and fold changes from C7 vs. the luminal cells. The genes represent the top 500 differentially expressed genes between C6 and luminal cells. **b** Volcano plot illustrating differential expression between C6 and C7, coloured dots represent significant genes with known function in lactation and immunity, dashed lines highlight the *P*-value threshold of 0.01 and a log fold change of 1. *P*-values are adjusted for multiple testing using Benjamini–Hochberg. **c**, **d** Visualisation of expression difference for genes related to lactation (**c**) or immunity (**d**) for the six luminal clusters in NP and PI. Expression values correspond to normalised UMI counts. **e** Cells from PI projected on the diffusion map from the NP and G time point. Cells from NP and G are coloured in grey

(Fig. 4e, f, Supplementary Data 17, Supplementary Fig. 6). On the alveolar branch, we found two clusters of genes that increase in expression during differentiation, one that is switched on early and another that is activated later in the final stages of differentiation. The early cluster is enriched for genes involved in fatty-acid oxidation and lipid biosynthetic processes (e.g. *Lipa*, *Dbi* and *Lpl*, Fig. 4f). The second, smaller group of genes that is switched on during the late stages of differentiation was enriched for genes involved in fatty-acid transport and lipid homoeostasis (e.g. *Olah*, *Cd36*, Fig. 4f). Of note, milk genes vary in their time of activation, with caseins switched on earlier during differentiation, whereas *Wap*, *Glycam1* and *Lalba* were only expressed in the final stages. Genes involved in cell division and translation show a transient phase of upregulation during the alveolar differentiation process (e.g. *Aurka*, *Cenpa*, *Cdk6*), whereas genes regulating cell shape and morphogenesis of a branching epithelium are repressed early on (e.g. *Vdr* and *Areg*). During the process of differentiation towards hormone-sensing luminal cells, we identified two broad clusters of genes that gradually increase or decrease their expression. Genes with increasing expression are involved in hormone metabolic processes and the regulation of morphogenesis of an epithelium (e.g. *Vdr*, *Esr1*, *Pgr* and *Msx2*). Amongst the branch-specific transcription factors we found, for example, *Runx1*, *Tox2* and *Bhlhe41* (also known as *Sharp1*) to be transcribed during differentiation towards the hormone-sensing lineage (Fig. 4c) and *Elf5*, *Foxs1* and *Ehf* in the secretory lineage (Fig. 4d). *Runx1* is a known repressor of *Elf5* and its deletion has been shown to be deleterious for ductal morphogenesis[28]. The expression of the known progenitor master regulator *Elf5*[29, 30] is maintained and further increased during secretory differentiation suggesting that its transcriptional level is fine-tuned in luminal progenitors (Fig. 4d).

**Parity primes luminal progenitors towards the alveolar fate**. Our data illustrate how the cellular composition of the gland changes during the pregnancy cycle. The luminal compartment shifts from giving rise to mainly hormone sensing cells to producing alveolar, milk-producing cells during pregnancy and lactation. The basal compartment on the other hand differentiates to produce oxytocin-sensing, myoepithelial cells that enable duct contraction and milk secretion during lactation. At the end of lactation, the gland then reverts back to a stage that resembles the pre-pregnancy state. However, we found that in particular the post-parous luminal compartment differs from its nulliparous counterpart (Fig. 1b, c). Here the effect of parity is most pronounced for the luminal progenitor cell types (Fig. 1c), which show larger numbers of differentially expressed genes as compared to the differentiated cell types (Supplementary Fig. 7a). C6 and C7 both showed characteristics of luminal progenitor cells, but we identified C6 predominantly in the nulliparous gland and C7 predominantly in the post-parous gland. To ensure that C7 still represents the progenitor population in the post-parous gland, we identified genes that distinguish C6 from the rest of the luminal compartment to see if they are also characteristic for the proposed post-parity progenitor population C7. Indeed, we find genes that are differentially expressed between C6 and the rest of the luminal compartment to show the same trend between C7 and the luminal compartment (Fig. 5a). In a similar manner we can distinguish C7 from the rest of the PI gland in a principal component analysis (PCA) using the identified progenitor genes (Supplementary Fig. 7b, c). From this we conclude that C7 indeed represents the post-parity luminal progenitor population. Interestingly, genes that were upregulated in C7 compared to C6 were significantly enriched for pathways that are involved in the immune response and lactation (Fig. 5b; Supplementary Fig. 7d).

These included genes that play roles in various processes during lactation, e.g. milk-proteins (*Csn2*, *Lalba*), lipases (*Lipa*), proteins involved in milk secretion (*Xdh*, *Cd36*) and transcriptional regulators of lactation (*Cidea*)[31]. Of note, the genes of the casein locus (*Csn2*, *Csn1s1*, *Csn1s2a*, *Csn3*) have previously been reported to be upregulated in the parous gland, most likely due to changes in chromatin accessibility[32, 33]. However, it has not been shown before that this effect is confined to the progenitor population of the luminal compartment. Some of the genes that are involved in the immune response are known regulators of the involution process such as *Ctsc*, *Tgfb3* or *Mfge8*[34]. The upregulation of these genesets was also present in C1 (Hsp) and in some cases even the differentiated C3 (Hsd) cluster, yet the effect remained strongest in the progenitor cells (Fig. 5c, d). Finally, we compared the differentiation processes of the luminal compartment of the parous gland to the nulliparous gland by mapping the luminal cells from PI to the trajectory of NP and G cells. Interestingly, C1 and C3 generally maintain the position of their nulliparous counterpart in the differentiation hierarchy whereas the C7 cluster is stretched out from the origin down towards the alveolar branch, suggesting that these are biased towards the alveolar fate (Fig. 5e). This is in agreement with the overexpression of genes related to the production and secretion of milk. Together, the data suggest that luminal progenitor cells maintain memory of having undergone gestation and lactation. This memory could potentially prime progenitor cells towards the alveolar fate to facilitate alveologenesis in subsequent pregnancies.

## Discussion

We have reported here the use of single cell RNA sequencing to comprehensively map the transcriptomes of thousands of mammary epithelial cells across four developmental time points. Our analysis identified 15 clusters of epithelial cells, some of which are only present during specific developmental stage. This study provides a rich data set that can be mined online (see link in Methods) to identify marker genes and lineage specific promoters that can be used to trace populations of cells in vivo. We note, however, that only some of the clusters can be fully characterised by a single marker gene. Instead we argue that the epithelial cells —especially in the luminal compartment—should rather be conceptualised as being part of a continuous spectrum of differentiation as visualised in Figs. 3 and 4. This view highlights the plasticity of the tissue and might help to explain the tissue homoeostasis of the adult mammary gland[4]. In this study, we could not detect any cells in transition between the basal and luminal cells. Therefore, we have no evidence to support the contribution of a putative multipotent stem cell to the day-to-day homoeostasis of the gland. We do note, however, that cells from C15 (Prc) expresses high levels of the stem cell marker *Procr*[19] and high levels of the luminal progenitor marker *Notch3*[35] (Fig. 2c). Yet, our data does not support a central role of this cluster in the day-to-day homoeostasis of the gland. In addition, cells from C15 also express some but not all markers of pericytes and it thus cannot be excluded that these are non-epithelial cells (Supplementary Fig. 4f). Based on the gene expression data presented here, the luminal compartment appears to have one common progenitor population (Lp). Lp gives rise to intermediate states of either hormone-sensing or secretory cells that express differentiation markers as well as progenitor markers and appear to represent meta-stable states on the differentiation path from luminal progenitors towards fully differentiated cells. Furthermore, we characterised gene expression patterns along the differentiation hierarchy. Here we identified genes that are lineage-specific, thus enabling us to disentangle the transcriptional

events that regulate differentiation of the luminal compartment. By analysing the mammary gland at various stages of development, we were also able to describe the effects of parity at cellular resolution. We found that the luminal progenitor compartment undergoes lasting changes at the transcriptional level. This is especially interesting in the light of the protective effect of pregnancies against breast cancer and the role of luminal progenitors as cell of origin. The majority of the changes were related to pathways involved in immunity and lactation, suggesting that, in particular, the luminal progenitors maintain a memory of gestation and lactation. We assume that C7 overlaps at least partially with the previously described parity-induced mammary epithelial cells (PI-MECs)[36].

In summary, our data provide an unbiased view of mammary gland development. This approach helps support some previously formed hypotheses in the mammary gland field and describes differentiation processes at a high cellular resolution. The data set will be a useful resource for future studies that aim to understand the relationship of the different cell types in the gland and how breast cancer develops and progresses.

## Methods

**Animals**. All experimental animal work was performed in accordance to the Animals (Scientific Procedures) Act 1986, UK and approved by the Ethics Committee at the Sanger Institute. C57BL/6N mice were housed in individually ventilated cages under a 12:12 h light–dark cycle, with water and food available ad libitum. The experiment was set up to allow for all of the developmental time points to be collected and tissues to be processed at the same time. Mice were euthanized by terminal anaesthesia. Females were mated with studs and allowed to litter. Tissues were then harvested at gestation day 14.5 (G), lactation day 6 (L) and day 11 post natural weaning of the pups (PI). Tissue from NP females was harvested at 8 weeks of age. Two individual mice per developmental time-point were used in the study. The oestrus cycle stage of animals from the NP and PI timepoint were determined by vaginal smears (Supplementary Fig. 8).

**Mammary gland dissociation into single-cell suspension**. Lymph node divested mammary glands (excluding the cervical pair) were dissected from the mice and mechanically dissociated. The finely minced tissue was transferred to a digestion mix consisting of DMEM/F12 (Gibco) + 10 mM HEPES (Gibco) + collagenase (Roche) + 200 U ml$^{-1}$ hyaluronidase (Sigma) + gentamicin (Gibco) for 3 h at 37 °C and vortexed every 30 min. After the lysis of red blood cells in NH$_4$Cl, cells were briefly digested with warm 0.05% Trypsin-EDTA (Gibco), 5 mg ml$^{-1}$ dispase (Sigma) and 1 mg ml$^{-1}$ DNase (Sigma), and filtered through a cell strainer (BD Biosciences).

**Cell labelling followed by flow cytometry and sorting**. Single-cell suspensions were incubated in HF medium (Hank's balanced salt solution (Gibco) + 1% foetal bovine serum, Gibco) + 10% normal rat serum (Sigma) for 20 min on ice to pre-block before antibody staining. All antibody incubations were performed for 10 min on ice in HF media. Mammary cells were stained with the following primary antibodies: CD31-biotin (eBioscience, clone 390, 1 μg ml$^{-1}$, 1:500); CD45-biotin (eBioscience, clone 30F11, 1 μg ml$^{-1}$, 1:500); Ter119-biotin (eBioscience, clone Ter119, 1 μg ml$^{-1}$, 1:500) and EpCAM-PE (Biolegend, clone G8.8, 0.5 μg ml$^{-1}$, 1:500). Cells were then stained with streptavidin-PE-CF594 (BD-Biosciences, 0.4 μg ml$^{-1}$, 1:500). Propidium Iodide (PI, Sigma, 1 μg ml$^{-1}$, 1:1000) was used to detect dead cells. Cells were filtered through a cell strainer (Partec) before sorting. Sorting of cells was done using a SH800Z sorter (SONY). Single-stained control cells were used to perform compensation manually and unstained cells were used to set gates. Chip alignment and sorting calibration was performed with automatic setup beads (SONY) immediately prior to sorting. Doublets, dead cells and contaminating haematopoietic, endothelial and stromal cells were gated out and EpCAM-positive cells were sorted in LoBind microcentrifuges tubes (Eppendorf) with HF. After sorting, cells were spun down and resuspended in HF. Samples were manually counted using an improved Neubauer chamber and the cell concentration was normalised by addition of HF. Equal numbers of cells per sample were processed for scRNA library preparation. Samples were processed for library preparation within 9 h from tissue isolation.

**Library preparation and sequencing**. Library preparation was performed according to instruction in the 10× Chromium single-cell kit. The libraries were then pooled and sequenced across six lanes on a HiSeq2500.

**RNA-seq data processing**. Read processing was performed using the 10X Genomics workflow[17]. Briefly, the Cell Ranger Single-Cell Software Suite was used for demultiplexing, barcode assignment and UMI quantification (http://software.10xgenomics.com/single-cell/overview/welcome). The reads were aligned to the mm10 reference genome using a pre-built annotation package obtained from the 10X Genomics website. All lanes per sample were processed using the 'cell-ranger count' function. The output from different lanes was then aggregated using 'cellranger aggr' with −normalise set to 'none'.

**Quality control and pre-processing**. In total, the Cell Ranger software identified 25,806 barcodes that contained enough unique molecules to be considered as cells (4376 in NP, 6021 in G, 9603 in L and 5806 in PI). Libraries prepared from all time points showed high quality that was reproducible between the two biological replicates (Supplementary Figs. 2 and 3). We used the following metrics to flag poor-quality cells: number of genes detected, total number of unique molecular identifiers (UMIs) and percentage of molecules mapped to mitochondrial genes. Poor-quality cells were then identified by setting a threshold on the number of genes and number of UMIs that was defined as three median absolute deviations (MAD) below the median for each time-point, while requiring a minimum value of 1000 total molecules and 500 genes detected. This resulted in the following thresholds for total number of genes detected: 1042 for NP, 836 for G, 500 for L and 759 for PI; and the following for total number of molecules: 2012 in NP, 1479 in G, 1000 in L and 1379 in G. In addition, all cells with 5% or more of UMIs mapping to mitochondrial genes were defined as non-viable or apoptotic and removed from the analysis (Supplementary Fig. 2d). We finally also ensured that none of the reads in our data set derived from index swapping[37, 38]. For this, we excluded cells with barcodes that appeared in more than one sample (non-unique barcodes). The reasoning behind this being that any index swapped read between samples will also share the same cellular barcode. Exclusion of these cells hence offers protection against index swapping. This left us with a total number of 25,010 cells (4223 in NP, 5826 in G, 9319 in L and 5642 in P). Gene expression values were then normalised by size factors that were estimated using the "computeSumFactors" function in scran[39]. The log-transformed (log2(counts + 1)) counts of highly variable genes (HVGs) were used as features for dimensionality reduction and clustering. HVGs were identified by first fitting a mean-dependent trend to the gene-specific variances to all genes assuming that this trend is dominated by technical variance. This trend was then used to estimate the technical component of the variance and all genes with a biological component (the residual variance) of at least 0.5 and a Benjamini–Hochberg adjusted $P$ value smaller than or equal to 0.05 were defined as HVG. The t-SNE embedding was computed using the 'Rtsne' package with default settings and perplexity set to 50 (https://github.com/jkrijthe/Rtsne).

**Clustering**. Due to the large number of cells, we used a two-step approach to identify clusters of cells. In a first step, we clustered cells using a shared-nearest-neighbour graph (SNN graph)-based approach. This has the advantage of not requiring the computation of a distance matrix that is computationally prohibitive for large numbers of cells. The SNN graph was constructed using the 'buildSNNGraph' function from scran with the number of nearest neighbours set to 20[40]. Community-based clustering was then performed on the SNN graph by multi-level modularity optimisation using the 'cluster_louvain' function in igraph[41]. This identified a total of 13 clusters. Despite this, the majority of clusters still contained substructure as evident from the t-SNE (Supplementary Fig. 4a). To resolve the cellular heterogeneity further, we then applied agglomerative hierarchical clustering on each of the 13 clusters (Supplementary Fig. 4b). For this, we first computed the pairwise distances between cells based on Spearman's rank correlation of log-transformed HVG counts. The dissimilarity matrix was then used to perform hierarchical clustering with the 'hclust' function in R using average linkage. Clusters were defined using the 'cutree' function and the optimal $k$ was determined by maximising the gap statistic using the clusGap function from the cluster package[42]. With this we identified a total number of 21 clusters. As the last step we performed a post-hoc test as previously reported[43] by merging clusters with less than 10 differentially expressed genes (at a $P$-value threshold of 0.01 and minimum log fold change of 1). This resulted in combining clusters 7 and 8 (Supplementary Fig. 4a) in C14. Next, we flagged clusters that expressed clear markers of non-epithelial cells as 'contaminating cells' and removed them from the downstream analysis. Clusters C16 and C17 were tagged as immune cells based on the expression of Cd74, Cd72 and Cd54[44], C18 as fibroblasts based on the expression of collagens and fibronectin and C19 as endothelial cells based on the expression of Eng, S1pr1 and Emcn (Supplementary Fig. 4c–e). We retained C15 as it expressed markers of a previously described cell type in the gland[19], despite being positive for 2 out of 4 (Des and Cspg4) pericyte markers[45] (Supplementary Fig. 4f). The high number of cells loaded on the 10× increased the chance of doublet formation in the droplet encapsulation[17]. We thus identified clusters that appeared at low frequencies (less than the maximum doublet rate of 7%, Supplementary Fig. 9a) and that were highly similar to at least two other clusters present in the same sample ($\rho > 0.9$ of mean log2(counts + 1)). The only cluster that fulfilled this criterion was cluster C20, whose gene expression was strongly correlated with C13 and C8 (Supplementary Fig. 9b). In addition, the gene expression profile of this cluster is virtually identical to the average expression values of C13 and C8, supporting the hypothesis that it is indeed a mixture of the two clusters (Supplementary Fig. 9b). As C13 and C8 are the most prevalent clusters in these samples (Table 1), doublets would be expected to be enriched for a combination of the two.

Finally, C20 also belonged to the clusters with a high number of genes detected and a high number of unique molecules, which is also indicative for doublet clusters (Supplementary Fig. 9c, d). We hence excluded C20 from any downstream analysis.

**Differential expression analysis**. Differential gene expression analysis was performed using edgeR[46]. For pairwise comparisons between clusters genes with a mean expression level below 0.1 were removed from the analysis. A negative binomial generalised log-linear model was fitted to the remaining genes with the cluster assignments as covariate(s). The 'glmTreat' function was used to identify genes that have a significantly higher log fold change than 1 at an FDR of 0.01. The marker genes visualised in Fig. 2 were identified using the 'findMarkers' function in scran with default settings on genes that had a median expression of 1 in at least one cluster[40].

**Diffusion maps and pseudotime inference**. For inferring the differentiation trajectory, we used diffusion maps. First, we selected all cells from the NP and G time point (Fig. 3a) and detected the HVGs as described above. The log-transformed (log2(count + 1)) gene counts were then used to compute the diffusion components using the 'DiffusionMap' function (default parameters as in destiny[47]). In Fig. 3b, we then focused on the luminal compartment and recomputed the diffusion map based only on the luminal cells, using the aforementioned procedure. Notably, the structure inferred by the diffusion map algorithm was robust to the choice of features and down-sampling of cells (Supplementary Fig. 5b). The structure of a common origin and the two branches could also be inferred using Monocle with standard settings[48] (Supplementary Fig. 5a). For inferring the branches and pseudotime ordering, we defined the following three tips, the cell with the largest value for the second eigenvector (which was set as root) and the cells with the largest and smallest values for the first eigenvector (compare Fig. 4a).

**Pseudotime-dependent expression**. To identify genes whose expression was significantly associated with the pseudotime, we first fitted a natural cubic spline with three degrees of freedom to the log-transformed (log2(counts + 1)) expression data in each branch. A likelihood ratio test was then used to assess statistical significance of the fit compared to a null (pseudotime-independent) model. Genes with a Benjamini–Hochberg corrected $P$-value below 0.01 and a minimum change in log2(expression) along pseudotime of 0.5 were considered to be significantly pseudotime-dependent. We then used a heuristic definition of branch-specific expression instead of modelling the branch assignment explicitly. This was motivated as follows. We were interested in general expression trends of genes, i.e. increase or decrease along the differentiation towards one of the two cell types, rather than comparing the exact timing of gene activation/inactivation between the branches. Any approach trying to do the latter would have been complicated by the different cell densities along the branches, differences in branch length and by the difficulty of verifying any such hypotheses in vivo. Hence, we defined genes to be branch-specific when they were pseudotime-dependent in their expression in at least one of the two branches and when the gradient differed in signs between two branches. The gradient was determined as the coefficient of a linear model fit to the spline-smoothed expression values, which was set to 0 if the coefficient was not significantly different from 0 at alpha = 0.01. Consequently, the gradient of a gene could either be −1 (decreasing), 0 (flat) or 1 (increasing).

**Gene set enrichment analysis**. A gene set enrichment analysis based on gene-ontology (GO) terms was conducted to characterise various genesets in the analysis. The genes of interest were compared to all genes that were tested for differential expression using topGO with default settings[49].

**Data availability**. The authors declare that all data supporting the findings of this study are available within the article and its supplementary information files or from the corresponding author upon reasonable request. The RNA sequencing data have been deposited in the Gene Expression Omnibus (GEO) database under accession code GSE106273. Data can also be explored at http://marionilab.cruk.cam.ac.uk/mammaryGland. All computational analyses were performed in R (Version 3.4.1) using standard functions unless otherwise indicated. Code is available online at https://github.com/MarioniLab/MammaryGland.

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

## Acknowledgements

We thank the staff at Sanger Institute, Research Service Facility (RSF) for their assistance. We also thank Dr. Aaron T. Lun (CRUK CI) for helpful discussions and comments on the manuscript. K.B. is funded by a Cambridge Cancer Centre studentship. D.A. is funded by CRUK and Wellcome Trust. S.P. is funded by CRUK. J.C.M. is funded by CRUK and EMBL. W.T.K. is funded by a CRUK career establishment award (C47525/A17348), University of Cambridge and Magdalene College, Cambridge.

## Author contributions

K.B. performed the experiments and all the computational analyses. S.P. and K.B. setup and collected the mammary epithelial cells. M.G. and J.H. performed the 10X library production and sequencing. J.C.M., D.A., K.B. and W.T.K. conceptualised the study and wrote the manuscript.

## Additional information

**Competing interests:** The authors declare no competing financial interests.

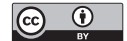

