## [Peer Review File · Nature Communications]

Reviewers' Comments:

Reviewer #1:

Remarks to the Author:

Bach and colleagues have performed single-cell RNA sequencing with mammary epithelial cells isolated at different developmental stages, i.e. virgin, pregnant, lactating and post-involution individuals. The study permitted to identify 9 cell clusters and establish hierarchical relationships between identified cell populations within luminal compartment. Notably, the data suggest a common origin of secretory and hormone-sensing lineages and well illustrate their segregation. Additionally, the study has provided a molecular characteristic of previously described parity-induced stem cell population. Much less basal cells were captured for the analyses, not permitting to establish a hierarchy within this compartment. The study is very interesting and important.

Specific comments

- Can it be excluded that Cluster 9 is a non-epithelial contamination? For instance, microvascular or perivascular cells? Cluster 9 cells, similar to myoepithelial cells, express some smooth muscle markers at low/medium level, however, they hardly express any epithelial marker.

- Samples from nulliparous and post-involution mice - NP1, NP2, PI1 and PI2. It is not mentioned at what stage of estrus cycle these mice were sacrificed, yet, it is known that gene expression is differentially regulated at different estrus stages. The data shown in Supplementary Figure 2a reveal significant differences in cell distribution between NP1 and NP2, as well as between PI1 and PI2. In pregnancy and lactation, hormonal levels should not differ significantly between individuals 1 and 2. Consistently, the samples 1 and 2 from pregnant and lactating mice, look similar within each stage - L1 and L2 samples, as well as G1 and G2 samples perfectly overlap in Supplementary Figure 2a. Ideally, cycling (NP and PI) mice should have been synchronized prior to cell sorting. This issue should be mentioned in the discussion.

- Why is Acta2 relatively high in some luminal clusters shown in Supplementary Figure 2c, – for instance, in Cluster 2, secretory luminal population? Is it “experimental noise”? Could the authors comment.

Page 7, lines 148-149. “... as well as transcription factors that have not previously been associated with luminal differentiation such as Creb5, Hey1...”. This statement is inexact. On the contrary, Hey1 is a well established marker of luminal progenitors (Bouras et al., 2008. Notch Signaling Regulates Mammary Stem Cell Function and Luminal Cell-Fate Commitment, Cell Stem Cell, 3: 429).

Page 9, lines 203-205. In contrast to authors' statement, it is not clear how the results of this work “might help to explain some of the conflicting results from lineage tracing studies”. The contradictions in lineage tracing data concern mostly the bipotency of basal stem cells, whilst, as the authors mention, the results of this work rather support the concept of lineage-restricted progenitor/stem cells in both compartments (Figure 2a).

Page 4, lines 87-88 “...from two independent mice.” What does “independent” stand for? Littermates?

Page 9, lines 189-190 “...luminal progenitor cells maintain memory of having undergone gestation and involution.” Maybe, “gestation and lactation”?

Reviewer #2:

Remarks to the Author:

The authors report on the first comprehensive single cell transcriptomic analysis of murine

mammary epithelia. This will be of significant interest to a large number of researchers interested in mammary development and neoplasia. The authors make several original claims, in particular the identification of two putative luminal progenitor populations and the appearance of lactational 'memory' in progenitors. The data analysis is well presented and clearly conducted, however I have concerns about sampling and reproducibility.

1. A major consideration with this paper is the use of Epcam to sort epithelial cells. Prior seminal publications (eg doi:10.1038/nm.2000) using human cells demonstrate an important contribution of Epcam negative epithelial cells to stem / progenitor activity in the mammary gland. What is the evidence that Epcam quantitatively recovers mouse mammary epithelia? The consequence of this methodology is the possible exclusion of subsets of mammary epithelial cells from the data capture.

2. Very few studies have used this system to sample diverse epithelial cell types, so it is not yet clear what sampling bias might be present in this method. The authors should compare the proportion of cell types represented in the single cell data with equivalents identified by flow cytometry at these developmental stages to address sampling bias.

3. The authors conducted analysis on biological replicates, however this is not described in the results section. This is an important consideration with a new technology and the variability between replicates should be appropriately quantitated and addressed in the results.

4. Given that the authors see this dataset as a resource, data should be made available for download.

5. Terminal End Buds (TEBs) are a critical mediator of pubertal development, with unique gene expression and function. TEBs start to disappear by the 8 weeks of age point used for nulliparous animals, so these cell types may not have been sampled. Do the authors observe TEBs at this time point?

Minor issues:

1. The link to the scripts on github is not operational

2. Was estrous stage controlled, measured or considered for the nulliparous mice? This will impact gene expression.

3. Please include a summary table of the number of cells captured from each animal, both total and in each cluster

4. The description of Fig 1 could use more cross referencing to known markers of each subset.

5. The colours of dots in each figure are very hard to discriminate, especially for those with diminished colour vision. Please try to choose a higher contrast palette or the use of patterns to distinguish groups.

6. Please label all axes, Eg. Fig A,D,E

7. Please conduct pathway and ontology analysis of the genes changing through pseudotime-described in lines 144-151.

8. C7 doesn't appear in Fig 2A. Please correct.

9. The genes referred to in lines 134-136 can't all be seen in Fig 1d.

Reviewer #3:

Remarks to the Author:

Bach et al. sorted mouse mammary epithelial cells (MECs) and used 10X Chromium system to perform single cell RNA-seq for individual MECs across four adult developmental stages: nulliparous (NP), mid-gestation (G), lactation (L), & post-weaning (PI). After filtration, they got single cell RNA-seq data for 3,471 individual cells (1,681 from NP, 609 from G, 604 from L, 605 from PI). They used unsupervised hierarchical clustering to analyze them and identified 9 clusters

of cells. Three of them are basal cells whereas the other six are luminal cells. Then they analyzed the detailed relation and differentiation trajectory of them. They further ordered the luminal cells by pseudotime analysis and analyzed the representative marker genes' expression dynamics. They showed that Cluster 4 cells are post-involution progenitor cells and the genes specifically expressed in them are enriched for pathways involved in the immune response and lactation. The work is interesting and gives new insights to the development of the mammary gland. However, some of the data are not of high enough quality and the authors need to confirm their results:

(1) The authors need to do independent experiments to confirm that the lactation stage cells really express so low number of genes. I am not convinced that an individual cell at L1 stage only express 400 genes. The authors should do single cell RNA-seq using SMART-seq2 protocol for all of these four stages of cells, probably five to ten single cells for each stage and check if lactation stage cells really express so low number of genes compared with the other three stages of cells.

(2) For Cluster 9, it seems that only 15 cells are there. And for Cluster 8, it seems that only 26 cells are there. Are they real clusters or just doublets? The authors should analyze further of these two clusters. For example, are C8 cells are really intermediate state of C2 & C5? Or they are merely doublets of C2+C5 cells? Probably the authors should capture all of the differentially expressed genes between C2 and C5 and check if C8 cells in general express these genes at 1/2 of the level of sum of C2+C5. Or maybe do some immunostaining to see if the cells double positive for C2 and C5 markers are really there.

(3) The authors should give the details of the filtration of the poor quality cells: exactly how many genes detected, how many UMI detected, what percentage of reads mapped to mitochondrial genes. Are they the same for all of the four stages of cells? Or different criteria for different stages of cells?

Dear Reviewers

Please find below our point-by-point response to your comments. One major point that was raised is the reproducibility of the analysis. Therefore, we have repeated the entire experiment and analysis to confirm the validity of our findings reported in the first version of the manuscript. In the repeat experiment we have sequenced approximately 23,184 cells (nearly 8x more cells), which allowed us to identify finer substructure within the data thus, further resolving the 9 clusters of mammary epithelial cells into 15 clusters. Reassuringly we have obtained the same relationships and differentiation trajectories between cells described in our initial submission. Our repeat dataset and analysis confirm and validate the main conclusions of our study. In particular we conclude that mammary epithelial cells should be viewed as a continuum especially in the luminal compartment and that most cell types within the gland cannot be defined by a handful of markers. In addition, even with 23,184 cells sequenced – we did not capture cells in transition between the basal and luminal compartments suggesting that both luminal and basal compartments are maintained by unipotent progenitor cells. This is in agreement with the majority of lineage tracing studies (1,2). Furthermore, we also confirmed that luminal progenitor cells maintain a memory of having gone through gestation and lactation. This is an interesting cell population to study, as there is a significant risk to developing breast cancer in the first five years post lactation and luminal progenitor's have been implicated as the cell of origin for aggressive types of breast cancer (3). The larger dataset of 23,184 cells will be a valuable resource for the mammary gland and breast cancer research community. The full raw data will be made available online via GEO and a user-friendly version through our website.

Reviewer #1

1. Can it be excluded that Cluster 9 is a non-epithelial contamination? For instance, microvascular or perivascular cells? Cluster 9 cells, similar to myoepithelial cells, express some smooth muscle markers at low/medium level, however, they hardly express any epithelial marker.

We thank the reviewer for the comments on the manuscript.

It is correct that there are similarities between the expression profile of Cluster 9 (called C15 in the revised dataset) and pericytes. However, there is also a significant overlap between C15 and the previously reported Procr^{+ve} cells (Wang D. et al. 2015 Nature). In the study by Wang et al. the authors demonstrate that Procr^{+ve} cells are epithelial cells with stem cell functions. Many of the upregulated genes in the Procr^{+ve} cells that were identified by the authors are also expressed at high levels in C15 (e.g. *Zeb1*, *Zeb2*, *Procr*, *Gng11*). In addition, the authors noted that Procr^{+ve} cells express only low levels of epithelial markers, which is the same for C15 in our dataset. Therefore, we believe C15 is a mammary epithelial cell cluster. However, we have now edited the text to clearly state that the expression profile of C15 in parts resembles that of pericytes (see Discussion, Page 10 lines 279-284).

2. Samples from nulliparous and post-involution mice - NP1, NP2, PI1 and PI2. It is not mentioned at what stage of estrus cycle these mice were sacrificed, yet, it is known that gene expression is differentially regulated at different estrus stages. The data shown in Supplementary Figure 2a reveal significant differences in cell distribution between NP1 and NP2, as well as between PI1 and PI2. In pregnancy and lactation, hormonal levels should not differ significantly between individuals 1 and 2. Consistently, the samples 1

and 2 from pregnant and lactating mice, look similar within each stage - L1 and L2 samples, as well as G1 and G2 samples perfectly overlap in Supplementary Figure 2a. Ideally, cycling (NP and PI) mice should have been synchronized prior to cell sorting. This issue should be mentioned in the discussion.

We have now included information on the estrus stage in the revised manuscript in Supplementary Figure 3 and 8. In the revised study both NP mice were in estrus while one of the PI mice was in estrus and the other in diestrus (Supplementary Figure 8). Due to the experimental setup and the need to process all eight animals for sequencing library preparation on the same day, it was not feasible to synchronise the PI mice. We found that there was a high level of correlation between all replicates (with no difference apparent for the PI mice), suggesting that the difference between estrus and diestrus is minimal (Supplementary Figure 3b). It is also important to note here that all of the identified clusters were composed of cells from more than one animal (Table 1).

3. Why is Acta2 relatively high in some luminal clusters shown in Supplementary Figure 2c, – for instance, in Cluster 2, secretory luminal population? Is it “experimental noise”?

This effect only occurs in cells from the lactation time-point, which has the highest number of myoepithelial cells (needed for the act of milk secretion) (Figure 2c, Table 1). Therefore, it is most likely background noise caused by ambient RNA from lysed cells.

4. Could the authors comment. Page 7, lines 148-149.”... as well as transcription factors that have not previously been associated with luminal differentiation such as Creb5, Hey1...”. This statement is inexact. On the contrary, Hey1 is a well-established marker of luminal progenitors (Bouras et al., 2008. Notch Signaling Regulates Mammary Stem Cell Function and Luminal Cell-Fate Commitment, Cell Stem Cell, 3: 429).

To address other reviewer comments, we repeated the experiment using a much larger set of cells. In this new, and much larger population, we did not observe a statistically significant change in the expression of Hey1 during pseudotime. Consequently, we have amended the text appropriately and thank the reviewer for pointing this out.

5. Page 9, lines 203-205. In contrast to authors’ statement, it is not clear how the results of this work “might help to explain some of the conflicting results from lineage tracing studies”. The contradictions in lineage tracing data concern mostly the bipotency of basal stem cells, whilst, as the authors mention, the results of this work rather support the concept of lineage-restricted progenitor/stem cells in both compartments (Figure 2a)

Our data from the repeat experiment with a much larger number of cells still supports the lineage-restricted model in both the luminal and basal compartments. We have now changed the phrasing of this sentence to reflect our conclusion.

6. Page 4, lines 87-88 “...from two independent mice.” What does “independent” stand for? Littermates?

For the repeat experiment all mice were not from the same litter except for the NP mice they were littermates.

7. Page 9, lines 189-190 “...luminal progenitor cells maintain memory of having undergone gestation and involution.” Maybe, “gestation and lactation”?

This is a fair point and we have edited the text accordingly.

Reviewer #2

1. A major consideration with this paper is the use of Epcam to sort epithelial cells. Prior seminal publications (eg doi:10.1038/nm.2000) using human cells demonstrate an important contribution of Epcam negative epithelial cells to stem / progenitor activity in the mammary gland. What is the evidence that Epcam quantitatively recovers mouse mammary epithelia? The consequence of this methodology is the possible exclusion of subsets of mammary epithelial cells from the data capture.

We thank the reviewer for their comments.

The use of Epcam to sort and isolate murine mammary epithelial cell is well established (Original research: Shehata M. et al. 2012, Breast cancer research, Prater M. et al. 2014, Nature Cell Biology and Giraddi R. et al. 2015, Nature Communications & Reviews describing the use of Epcam: Visvader J. 2009, Genes and Development, Visvader J. and Stingl J. 2014, Genes and Development and Oakes S. et al 2014, Cell Mol Life Sci.). Based on these studies Epcam is a comprehensive marker of murine mammary epithelial cells.

2. Very few studies have used this system to sample diverse epithelial cell types, so it is not yet clear what sampling bias might be present in this method. The authors should compare the proportion of cell types represented in the single cell data with equivalents identified by flow cytometry at these developmental stages to address sampling bias.

At present there is not a sufficient repertoire of cell surface markers that allow isolation and identification of all the epithelial cell populations described in our dataset by FACS. In addition, one of the key findings from our study is that the use of a handful of markers to define the various mammary epithelial compartments is not appropriate given the continuum of differentiation states observed in the gland. With regards to the point about sampling – this is one of the main reasons why we decided to repeat the experiment and capture more cells for sequencing. The repeat experiment validated and confirmed the same diversity of mammary epithelial cells observed in the first dataset albeit at a higher resolution.

3. The authors conducted analysis on biological replicates, however this is not described in the results section. This is an important consideration with a new technology and the variability between replicates should be appropriately quantitated and addressed in the results.

We now comment on the reproducibility of the replicates in the results section and we include more detailed analysis of the replicates in Supplementary Figure 2, 3 and 8.

4. Given that the authors see this dataset as a resource, data should be made available for download.

Indeed the raw data will be available to download via GEO and a user-friendly version through our website (<http://marionilab.cruk.cam.ac.uk/mammaryGland>).

5. Terminal End Buds (TEBs) are a critical mediator of pubertal development, with unique gene expression and function. TEBs start to disappear by the 8 weeks of age point used for nulliparous animals, so these cell types may not have been sampled. Do the authors observe TEBs at this time point?

In this study we focused on post-pubertal mammary epithelial development. The reviewer is correct that the majority of TEBs should not be present post-puberty (8 weeks) and we did not detect a NP specific cluster expressing TEB associated genes.

Minor issues:

1. The link to the scripts on github is not operational

Apologies. We have now amended the link and it is functional.

2. Was estrous stage controlled, measured or considered for the nulliparous mice? This will impact gene expression.

Yes, the mice were checked and this data is now included in the manuscript. See response to question 2 from reviewer #1.

3. Please include a summary table of the number of cells captured from each animal, both total and in each cluster

This is now included as Supplementary Figure 2a and Table 1.

4. The description of Fig 1 could use more cross referencing to known markers of each subset.

We have now split Figure 1 into two separate figures (Figure 1 and 2) and included more cross referencing as suggested to better highlight the different clusters of cells and their putative identities.

5. The colours of dots in each figure are very hard to discriminate, especially for those with diminished colour vision. Please try to choose a higher contrast palette or the use of patterns to distinguish groups.

We have changed the colour palette to one with a higher degree of contrast.

6. Please label all axes, Eg. Fig A,D,E

Fixed.

7. Please conduct pathway and ontology analysis of the genes changing through pseudotime- described in lines 144-151.

We have performed clustering on the pseudotime dependent gene expression and gene ontology analysis in the revised manuscript, which can be found in Figure 4 and Supplementary Figure 6.

8. C7 doesn't appear in Fig 2A. Please correct.

We have now included a different aspect of the cellular trajectory. However, given the number of cells on the plot it is unavoidable that cells will be masked by other cells.

9. The genes referred to in lines 134-136 can't all be seen in Fig 1d.

This is now fixed. In addition, all differentially expressed genes are listed in Supplementary Table 1.

Reviewer #3

1. The authors need to do independent experiments to confirm that the lactation stage cells really express so low number of genes. I am not convinced that an individual cell at L1 stage only express 400 genes. The authors should did single cell RNA-seq using SMART-seq2 protocol for all of these four stages of cells, probably five to ten single cells for each stage and check if lactation stage cells really express so low number of genes compared with the other three stages of cells.

Following the reviewer's suggestion, we explored this issue in more detail. Upon re-analysing the dataset reported in the first submission we identified a potential technical issue related to index swapping (4), which could potentially explain the lower number of expressed genes in the lactation sample. To determine whether this was indeed the case, we repeated the experiment using independent mice, generating many more cells at each stage. While the overall results regarding the identity of cell types, the independence of the compartments and

the sets of genes upregulated specifically on different trajectories remained unaltered, the number of genes identified as expressed in the lactation population of cells was much more similar to the other stages. Consequently, this observation no longer holds and we have removed it from the text (where we also focus exclusively on the new dataset).

2. For Cluster 9, it seems that only 15 cells are there. And for Cluster 8, it seems that only 26 cells are there. Are they real clusters or just doublets? The authors should analyze further of these two clusters. For example, are C8 cells are really intermediate state of C2 & C5? Or they are merely doublets of C2+C5 cells? Probably the authors should capture all of the differentially expressed genes between C2 and C5 and check if C8 cells in general express these genes at 1/2 of the level of sum of C2+C5. Or maybe do some immunostaining to see if the cells double positive for C2 and C5 markers are really there.

We agree with the reviewer that doublets are an important confounding factor in single-cell studies. For this reason, we have now incorporated a step in our cluster analysis to identify potential doublet clusters. We excluded clusters that fulfil the following two criteria. First, the cluster needs to appear at frequencies below the maximum expected doublet frequency in all samples where it was detected in order for it to be considered a potential doublet. Second, this cluster also needs to be highly correlated with at least two other clusters from the same sample(s). The only cluster that fulfilled these criteria is C20, which appeared to be a mixture for C13 and C8 (Supplementary Figure 9). Neither C9 (now C15) nor C8 (now C10) fulfilled these criteria. This step is also mentioned in the materials and methods section of the main text.

3. The authors should give the details of the filtration of the poor quality cells: exactly how many genes detected, how many UMI detected, what percentage of reads mapped to mitochondrial genes. Are they the same for all of the four stages of cells? Or different criteria for different stages of cells?

We used the following metrics to flag poor quality cells: number of genes detected, total number of unique molecular identifiers (UMIs) and percentage of molecules mapped to mitochondrial genes. Poor quality cells were then identified by setting a threshold on the number of genes and number of UMIs that was defined as three median absolute deviations (MAD) below the median for each time-point, while requiring a minimum value of 1,000 total molecules and 500 genes detected. This resulted in the following thresholds for total number of genes detected: 1,042 for NP, 836 for G, 500 for L and 759 for PI; and the following for total number of molecules: 2,012 in NP, 1,479 in G, 1,000 in L and 1,379 in G. In addition, all cells with 5% or more of UMIs mapping to mitochondrial genes were defined as non-viable or apoptotic and removed from the analysis (Supplementary Figure 2d). This is now also mentioned in the materials and methods section.

References

1. Van Keymeulen, A. *et al.* (2011). Distinct stem cells contribute to mammary gland development and maintenance. *Nature*, 479(7372), 189–193.
2. Davis, F. M. *et al.* (2016). Single-cell lineage tracing in the mammary gland reveals stochastic clonal dispersion of stem/progenitor cell progeny. *Nature Communications*, 7

3. Molyneux, G. *et al.* (2010). BRCA1 basal-like breast cancers originate from luminal epithelial progenitors and not from basal stem cells. *Cell Stem Cell*, 7(3), 403–417.
4. Sinha, R. *et al.* (2017) Index switching causes ‘spreading-of-signal’ among multiplexed samples in Illumina HiSeq 4000 DNA sequencing. *bioRxiv* 125724

Reviewers' Comments:

Reviewer #1:

Remarks to the Author:

The revised version is significantly improved. The author have addressed the criticism. Moreover, the number of the analyzed single cells has been increased by several folds permitting to refine clustering within cell types defined in the original version.

Reviewer #2:

Remarks to the Author:

I am satisfied that the authors have addressed my questions.

Reviewer #3:

Remarks to the Author:

The authors have addressed all of my concerns and it can be accepted now.